# Quality of Clinical Guidelines on Oral Care for Children with Special Healthcare Needs: A Systematic Review

**DOI:** 10.3390/ijerph20031686

**Published:** 2023-01-17

**Authors:** Hamdan Alamri, Falah R. Alshammari, Abdullah Bin Rahmah, Marwan Aljohani

**Affiliations:** 1Department of Preventive Dental Sciences, College of Dentistry, Majmaah University, Al-Majmaah 11952, Saudi Arabia; 2Dental Public Health and Community Dentistry, College of Dentistry, University of Ha’il, Ha’il 55476, Saudi Arabia; 3Department of Periodontics and Community Dentistry, College of Dentistry, King Saud University, Riyadh 11362, Saudi Arabia; 4Department of Oral and Maxillofacial Surgery, College of Dentistry, Taibah University, Medina 42353, Saudi Arabia

**Keywords:** evidence-based dentistry/healthcare, guidelines, the AGREE II tool, systemic review, children with disability, children with special healthcare needs (SHCNs)

## Abstract

Robust evidence-based guidelines are important in everyday clinical practice, especially when delivering and managing oral care needs to a vulnerable group such as children with special healthcare needs (SHCNs). Methods: To assess the quality of guidelines on the management of oral care for children with special healthcare needs (SHCNs) and to find appropriate guidelines, an electronic search of MEDLINE Ovid was carried out alongside an additional search of common guideline websites. The AGREE II tool was used to assess the quality of the guidelines. Assessment was undertaken independently by three assessors. Furthermore, the underlying evidence used to formulate recommendations in the identified guidelines was qualitatively assessed. Results: There were nine guidelines, with 41 recommendations, that met the eligibility criteria. The quality of the guidelines was generally found to be poor. Only one guideline was assessed as “recommended” by the assessors, based on the quality of the methods, the reporting, or both. Only 2 of the 41 sets of recommendations, made across the nine guidelines, were judged to be valid and based on a rigorous systematic review of the evidence. Conclusions: The current state of guidelines on oral care management for children with special healthcare needs (SHCNs) is, on the whole, of very low quality. The scientific community should work together to enhance the quality and strength of the current clinical guidelines and to ensure that they are trustworthy prior to implementation.

## 1. Introduction

Children are categorized as having a special healthcare need if they are living with a health condition that limits their ability to function in one or multiple aspects. These can include physical mobility, cognitive or sensory functions, memory, or self-control, among a variety of other aspects [1]. Children with special needs have been defined as “young people experiencing serious and persistent physical, psychological and/or social problems” [2,3]. Impairments as well as activity limitations and restrictions all fall under the term “disability” [4]. Some children live with a single disability or need; however, in more severe cases children may suffer from a mixture of physical, developmental, cognitive, and affective limitations [5]. Accordingly, children with SHCNs may have significant restrictions in oral hygiene performance because of their intellectual, sensory, and motor needs, and therefore are more likely to suffer oral health issues than other children [6,7,8,9,10,11].

Deterioration of oral health among children with SHCNs can, in part, be attributed to various underlying factors such as enamel irregularities, craniofacial birth defects, impaired salivary function, malocclusion, periodontal disease, and more frequent oral infections [12,13]. Additional factors include an over-dependence on a healthcare professional for regular oral hygiene, frequent use of medicines that have high sugar content, preference for foods rich in carbohydrate, oral aversions, inadequate oral cavity care, and a liquid or semi-liquid diet [14].

Oral health conditions for children with special healthcare needs (SHCNs) are often poorer when compared with healthy children [11,14,15]. Oral health is a very important aspect of general health and, in order to maintain overall health, it is important to consider maintaining good oral health [16]. Specifically, children with SHCNs face many everyday challenges or problems in maintaining good oral health [17,18]. This group of children often present additional challenges, such as complex medical conditions and behavioral issues, which can result in more difficulties in obtaining appropriate dental care [19]. Many research studies have reported that people with SHCNs are at a higher risk of dental diseases compared with others and are also more likely to have untreated or unmet dental or oral needs [18,20,21].

The receipt of optimal oral healthcare is fundamental for children with SHCNs as it can provide them with confidence, thereby allowing them to reach their full potential as well as take part in society. Providing children with SHCNs with appropriate oral healthcare necessitates additional treatment and care from all health workers, where they take into consideration the physical and medical limitations of the child [11]. One important aspect of achieving reasonable oral healthcare for children with SHCNs is the development of clinical guidelines to address the needs of this population in a dental setting. Moreover, guidelines are intended to assist oral health providers and clinicians in planning, as well as in delivering, the highest-quality healthcare.

Guidelines are mainly aimed at providing oral health practitioners and clinicians with a set of explicit recommendations and suggestions regarding how to deal with certain situations, as well as to minimize the use of harmful, ineffective, or unnecessary actions or measures. Those developed recommendations should be the result of a long process of consideration of valid and up-to-date evidence for the correct management and treatment of the dental patient. By reducing inconsistencies and variations between clinical practice and scientific evidence, clinical guidelines can enhance the quality of dental care that is delivered to the patients. Guidelines should be rigorously developed and precise, ensuring that any resulting recommendations are clear and reproducible [22].

Several factors can limit the usefulness and/or the applicability of clinical guidelines. These include the guideline development methodology, the availability of sound research evidence, the uniqueness of individuals, and how the research findings can be generalized. On that basis, guidelines for the management of oral health in children with SHCNs should be appraised as it is very important to identify and analyze any potential flaws in their development [23,24]. Indeed, different appraisal tools have been developed to determine the quality of clinical guidelines [22].

The AGREE II tool is an instrument that is used widely for the assessment of transparency, accuracy, and the methodological rigor of guideline development and has been tested for its reliability as well as validity. The AGREE II tool is to be applied or used by guideline developers to properly guide their work and to evaluate or assess the quality of their methodology (the AGREE Collaboration). In addition, it also offers a preferred methodological framework for the development of content for the guideline. This instrument has been designed and developed to assist in the standardized and objective appraisals of guidelines, as not all clinical guidelines are developed or conducted transparently and rigorously [22]. The aim of the current study was to critically appraise the most recent guidelines on oral healthcare management for children with SHCNs in a pediatric dentistry setting, using the AGREE II tool. A secondary objective was to evaluate the underpinning evidence behind the recommendations listed in the appraised guidelines.

## 2. Materials and Methods

A systematic review was undertaken to identify clinical guidelines and recommendations focusing on oral healthcare and treatment of children with SHCNs in dentistry. The AGREE II instrument was used in this study to critically evaluate clinical guidelines for the treatment and care of children with SHCNs.

### 2.1. Eligibility Criteria

We included guidelines that met all of the following inclusion criteria:Guidelines published in the English language and within the past ten years;Only guidelines listed for specific or clinical circumstances or conditions for children and with SHCNs related to oral healthcare were included;Only guidelines intended for or directed towards practitioners to assist in handling, as well as treating, oral health for children with SHCN in pediatric dentistry were included.

The exclusion criteria for the guidelines were:Guidelines that have been superseded or modified by a more recent version carried out by the same group of guideline developers;Guidelines that were reproduced versions of previous or current guidelines and have not been reproduced with amendment.

### 2.2. Information Sources and Search Strategy

A search of the MEDLINE Ovid database was conducted using the search terms that are shown in Appendix A. Additionally, eight guideline developer websites were searched thoroughly to find clinical guidelines for children with SHCNs in dentistry Appendix A.

### 2.3. Selection of Guidelines

One assessor examined and evaluated the titles as well as the abstracts of each of the retrieved records to determine which ones were appropriate and should be selected as per the eligibility criteria. If the retrieved records were not clear from the titles and abstracts, a full-text copy was checked for any potential eligibility.

### 2.4. Guideline Assessment Using the AGREE II Instrument

The AGREE II instrument aims to provide a framework to assess or analyze the quality of clinical guidelines; to provide a procedural and systematic strategy for guideline development; to inform or explain the preferred content and the reporting of the content in clinical guidelines. The AGREE II instrument is comprises 23 key items within six domains, with two additional global rating items. The six domains are presented in Table 1.

All 23 items, across the six domains, are rated on a 7-point scale, where a score was assigned based on reporting quality and completeness. A score between two and six was given when the reports of the item failed to address all the item considerations or full criteria (score of 1 means (*strongly disagree*) vs. score of 7 (*strongly agree*)).

All guidelines meeting the inclusion criteria were rated independently by three appraisers to increase the reliability and accuracy of the assessment. A pilot exercise was undertaken by the assessors to establish consistency in the scoring process. It is important to note that a level of subjectivity is required in rating the guidelines. An overall assessment using the AGREE II instrument involves two additional items: the first was to ask the appraiser to judge the overall quality of each of the reviewed guidelines on a 1 to 7 scale and the second was to ask the appraiser if they would recommend the use of the guideline, with *yes*, *yes with modifications*, or *no*.

### 2.5. Calculating the Domain Scores

A quality score was calculated for each of the six domains in AGREE II by totaling the scores of the domain’s items and by scaling the total score as a percentage of that domain’s maximum possible score, as described in the AGREE II user’s manual. All the six domain scores were independent and therefore were not aggregated into one quality score.

## 3. Results

From the search, a total of 314 records were identified. From an initial screening, 89 guidelines were identified for further review, of which 9 guidelines were deemed eligible for inclusion (Figure 1). These nine guidelines were critically appraised using the AGREE II instrument. The overall results are presented below in Table 2 and Table 3. Furthermore, the median and the range scores for each domain were reported in Table 4 to summarize the overall result of the included clinical guidelines.

The assessors’ ratings provided in Table 4 suggest that according to the AGREE II instrument, guidelines on oral healthcare for children with SHCNs are typically of low quality. This was perhaps best demonstrated by the fact that all assessors would only recommend one of the nine guidelines. Following the AGREE II user’s manual, there is no guidance concerning thresholds for high or low quality and that the end-user should use their judgment for interpretation. For all the domains, a cut-off score of >60%, which has been used in the literature before, was used in this study to indicate a high-quality guideline [34].

The first domain, “Scope and Purpose”, generally scored well, with the median score across the nine guidelines of 80% (range 67 to 85%). All the included guidelines scored greater than 60%. However, clinical guidelines are likely to score high in this domain, as this is a fundamental part of the guideline development process. Meanwhile, the second domain, “Stakeholder Involvement”, generally fared less well, with a median score of 28% (range 17 to 83%) and only two guidelines scored greater than 60% [27,33]. Furthermore, in this domain, the views and preferences of the target population (item 5) had not been sought or discussed across the guidelines, except for the two-aforementioned guidelines.

The third domain, “Rigor of Development”, was judged as poor, with a median score of 28% (range 3 to 88%) and with only one guideline that scored > 60% (Crystal et al., 2017). All the included guidelines had noticeably low scores across all items in the “Rigor of Development” domain, especially items 13 (external review of the guidelines) and 14 (guideline updating procedure provided). In addition, we judged the fourth domain, “Clarity of Presentation”, to be poor for four guidelines, scoring <60%, with a median score of 63% (range 35% to 85%). Furthermore, a high score in this domain by presenting the recommendations in a clear format should not be too difficult to achieve; however, four guidelines fell short in this area [25,26,28,31]. For the fifth domain, “Applicability”, the majority of guidelines were judged to be poor, with a median score of 15% (range 0% to 60%) and only with one guideline scoring > 60% Crystal et al., (2017) [33]. Even though the guidelines were developed largely by professional organizations such as the American Academy of Pediatric Dentistry AAPD, all the included guidelines, except for Crystal et al., (2017), failed to consider and report guideline implementation [33]. The final domain, “Editorial Independence”, scored extremely low, with a score of zero across all guidelines. Looking at the scores of this domain, no clinical guidelines included explicit statements regarding independence or any competing interests arising during the formulation or development of their recommendations.

### Quality of Underlying Evidence

When inadequate focus is applied to the underpinning evidence in guidelines, there is a chance that incorrect recommendations may be given, subsequently leading to clinicians’ performance being potentially less than optimal for their patients. Recognizing the relevance and quality of the underpinning evidence can of course mitigate such negative outcomes. The majority of guidelines do not take into account the extent to which evidence can be generalized for people, interventions, and outcomes [35,36]. Whilst applying the AGREE II instrument, it became clear that at no point were queries raised concerning the quality of the evidence that underpinned the guidelines. To complete this study on guidelines for the management of oral care for children with SHCNs, it was important to examine/appraise the underpinning evidence in the included guidelines.

The main researcher, with a background in the topic covered in the included guidelines, documented the recommendations and underlying evidence cited by each guideline. Striving for an objective review, all guidelines were thoroughly investigated to determine whether high-quality evidence (systematic reviews) had been used as the foundation for any recommendations by appraising the references presented. References to systematic reviews were examined independently and in duplicate to determine the quality and the relevance of the underlining evidence.

Overall, 41 sets of recommendations were reviewed in the included guidelines. Naturally, some guidelines dealt with multiple conditions rather than just one. The most frequently mentioned issue was the use of general anesthesia, sedation, and behavior management of children with SHCNs (in seven of the nine guidelines). This was followed by the use of clinical holding and the use of fluoride. Among the recommendations reviewed, only two were deemed to be sufficiently supported by a high-quality systematic review. Furthermore, the recommendations in just two of the reviewed guidelines had been allocated a grade/level that the remaining guidelines failing to do (Table 4).

**Table 4 ijerph-20-01686-t004:** Baseline characteristics for the guidelines.

Guideline	Population	Condition or Type of Intervention (Prevention, Diagnostic Test, and Treatment)	Grade or Level of Recommendation(s)	Recommendation(s) Supported by High-Quality Evidence?	Comment
Guideline on behavior guidance for the pediatric dental patient [25]	Pediatric dental patients (behavior guidance)	Behavior management	None stated	No	The recommendations listed were based on a narrative review of the literature (not a systematic review). This guideline only referenced one systematic review and it is not related to the final recommendation.
Clinical holding skills for dental services [26]	Specific to children with disabilities	Clinical framework	None stated	No	This is a framework for using restrictive interventions. Not clear how the evidence was searched and how recommendations were formulated. This guideline does not cite any systematic reviews.
Guideline on use of anesthesia personnel in the administration of office-based deep sedation/general anesthesia to the pediatric dental patient [28]	Pediatric dental patients	Regulatory measures for pharmacological management (deep sedation/general anesthesia in clinic)	None stated	No	The recommendations listed were based on narrative reviews of the literature. Very vague recommendations that do not reference any systematic review. This publication aims to provide a list of regulatory measures for using pharmacologic behavior guidance in dental settings.
Clinical guidelines and integrated care pathways for the oral healthcare of people with learning disabilities [27]	Specific to individuals with disabilities	Prevention of oral diseases and the maintenance of good oral health	SIGN grading levels A, B, or C. Grade A (at least one randomized controlled trial); B (conducted clinical studies but no randomized clinical trials on the topic of recommendation); C (requires evidence from expert committee reports or opinions and/or clinical experience of respected authorities).	Unclear	No systematic review(s) were used to inform this guideline. Recommendations were listed without referencing the individual body of evidence. (Recommendations were labeled using SIGN grading levels but not linked to the specific body of evidence.)
Guideline on management of dental patients with special healthcare needs [29]	Specific to individuals with SHCNs	Management of oral healthcare needs for individual with SHCNs	None stated	No	The recommendations listed were based on a narrative review of the literature (not a true systematic review). The guideline only references one systematic review and it is not related to the final recommendations. The recommendations are vague and general with only four of the recommendations related to children with SHCNs.
Guidelines for monitoring and management of pediatric patients before, during, and after sedation for diagnostic and therapeutic procedures [30]	Pediatric dental patients	Safety and management measures of sedation	None stated	No	The recommendations listed were based on a narrative review of the literature (not a true systematic review). This publication aims to provide an updated statement to unify the guidelines for sedation, however the recommendations are vague and general. The guideline did not reference any systematic review.
Guideline on use of nitrous oxide for pediatric dental patients [31]	Pediatric dental patients	Practical aspects of delivering nitrous oxide	None stated	No	The recommendations listed were based on a narrative review of the literature (not a true systematic review). Although the guideline specifically mentions children with SHCNs as an indication to use this treatment, there were no specific recommendation or measures that listed this population. Very vague recommendations that overlap with the previous guidelines referenced (documentation, monitoring, facilities/personnel/equipment). The guideline dose not reference any systematic review.
Guideline on prescribing dental radiographs for infants, children, adolescents, and persons with special healthcare needs [32]	General and specific to patients with SHCNs	Timing and prescribing of radiographs	None stated	No	The recommendations listed were based on a narrative review of the literature (not a true systematic review). Although the guideline specifically mentions patients with SHCN, there was no specific recommendation for this population. The guideline does not reference any systematic review.
Use of silver diamine fluoride for dental caries management in children and adolescents, including those with special healthcare needs [33]	General and specific to children with SHCNs	Treatment and application of silver diamine fluoride	(GRADE) approach was used	Reference one systematic review in 2016 with 4 RCTs	The guideline is largely informed by an existing systematic review (Zhao, 2016 [37]). It is also informed by other guidelines, clinical studies, and expert opinion. (Authors conclude that there is very low-quality evidence (GRADE), none of which included children with children with SHCNs.)

## 4. Discussion

It is important to acknowledge that certain limitations in the review process may have had an impact on this guideline appraisal work. Firstly, it is important to emphasize that, despite our best efforts, this was not a comprehensive or exhaustive assessment of clinical guidelines in relation to oral healthcare for children with SHCNs. Some clinical guidelines may have been missed as a result of human error or due to the electronic search strategy. Some clinical guidelines do not include children with SHCNs in their headings or abstracts as a potential beneficiary, which makes it challenging to identify potentially eligible guidelines. Further, guidelines are rarely indexed and are not always published, making their identification and retrieval difficult. For guidelines to have any chance of improving clinical practice, as is their primary intention, they need to be easily accessible.

The AGREE II tool has certain limitations in the search for quality evidence that underlines the final set of recommendations in a guideline. The AGREE II instrument cannot highlight such a deficiency, as the focus on methodological issues that are linked to guideline development is insufficient to ensure that the final set of recommendations is valid. Hence, this tool is used to evaluate the process of guideline development and the way it is reported [38]. Furthermore, by working through the appraised guidelines it became apparent that the supporting evidence underpinning the final recommendations was based on evidence of variable quality. This is due to the fact that most guidelines failed to incorporate high quality and valid evidence such as a systematic review in some or all the final sets of recommendation; thus, this requires further investigation.

Using the AGREE II tool, most guidelines had the highest scores in the “Scope and Purpose” domain. Indeed, this had been anticipated as this domain comprises fundamental components of a guideline that cannot be easily neglected, such as the target population, the health questions that are being addressed, and the objectives of the guidelines. Therefore, guideline developers usually focus more on these parts of the guideline when developing their papers.

Participation in or co-development of guidelines was especially limited, as indicated in the “Stakeholder Involvement” domain. Only two clinical guidelines included members belonging to other professional groups as developers of guidelines [27,33]. Furthermore, the views of the patients were not taken into consideration in the development of the clinical guidelines, but rather the patients/public participated as external reviewers after the guidelines had been developed, except for the two aforementioned guidelines. This speaks poorly of the developers of guidelines in the field, as it is very important to consider the views of those for whom the guideline is developed, which may also help later on in the successful implementation of the guidelines [39].

The “Rigor of Development” domain is a strong indicator of the quality of clinical guidelines as it represents the methodology part as compared with other domains. A high score in this domain indicates that the guideline has been developed with minimum bias and is based on evidence, whereas a low score, on the other hand, indicates potentially serious problems in the methodological approach that is used for the development of guidelines [40]. However, all the included guidelines, except for one, had incomplete or entirely missing methodological details, such as the presence of external inputs by experts prior to the guideline publications or including a timeline for the guideline updating process [33]. Furthermore, guidelines need to reflect the current literature and provide a revision or updated procedure in the guideline’s development, which is a fundamental step in the identification of new evidence that may impact existing recommendations. Guidelines can become outdated as new evidence emerges and therefore developers should prospectively determine when and how they will update a guideline. In the appraised guidelines, most were missing a statement regarding the process for updating the guideline process, except for two [27,33].

Guideline developers in pediatric dentistry make use of rigorous methods that include the participation and opinions of all the relevant stakeholders. It is also important that clinical guidelines should have a detailed and structured approach in order to assess the quality and to evaluate the strength of evidence that they use to support their final recommendations. Such clinical guidelines must also have a clear link between the final recommendations and the evidence. Clinical guidelines should ideally be transparent with regard to the methodological strategy used, with the support of evidence that helps to produce the final recommendations of their work [38].

In general, it is very much evident that all the included clinical guidelines, except for one, failed to perform well with regard to the “Stakeholder Involvement” and the “Rigor of Development” domains [33]. The primary focus of these domains is based on the methodological part of guideline development. The results or findings of these domains require correct attention from guideline developers, as the gathering of evidence, along with meticulous interpretation, means that the accuracy and quality of each included study must be assessed and appraised individually step by step. This is vital and necessary, as it is unsatisfactory to assess based only on the study design (i.e., meta-analyses or RCT) to be high-rated evidence, as such studies can themselves have sources of bias or methodological issues. Furthermore, some clinical guidelines have linked or connected similar recommendations to a broader or even completely different body of evidence. In the same way, although some of the clinical guidelines made use of grading systems to evaluate the evidence strength or quality, all the appraisal guidelines bar one did not evaluate or analyze the individual quality or strength of the studies [33].

In the fourth domain, “Clarity of Presentation”, four of the included guidelines score poorly, as they failed to present the recommendations in a clear format. Even if the underlining evidence is unclear, the uncertainty should be highlighted as it is. Furthermore, in the “Applicability” domain, almost all the guidelines attained a low score except for one [33]. The primary reason for this was due to the methods required for the successful implementation of the clinical guidelines that were not reported clearly and precisely. Furthermore, the detailed and descriptive report of the barriers, as well as the facilitators and implications of applying the recommendations, were missing in the included guidelines. Further, a lack of economic perspective was observed in the evaluated guidelines. This is very much relevant, as any clinical decision has implications on benefits as well as costs to patients and many other agents such as health suppliers and society.

The clinical guidelines must include a clear and detailed statement showing that the final recommendations have not been influenced by the interests or views of the funding body, and they must also include a clearly stated description of any sector’s contribution to the development of guidelines. Even though the bodies that fund these guidelines are mainly academic institutions and governmental agencies, it was noted by the appraisers that additional details on the purpose and role of these funding bodies or agencies were missing in the content of the guidelines. In order to accomplish the criteria of the AGREE II tool, there needs to be a detailed statement that shows how the interests of the funding body have not influenced or affected the final recommendations. Simultaneously, the authors of all the guidelines should offer a disclosure or report of all competing interests. Nonetheless, as per the reviewers or assessors, this information has not been reported adequately in all the clinical guidelines. This aspect is important, as it is clear that conflicts of interest among guideline authors are very common and the quality of final recommendations may be affected as a result of this. It is therefore important to pay attention to this domain’s quality [41].

The second part of the research (Table 4) examined the extent to which the recommendations had been supported by high-quality evidence (i.e., systematic review). The guidelines in our study, of which there were nine, varied in length from 4 pages to 99 and cited anywhere from 5 references to 259. The findings from this appraisal were concerning. Across every recommendation reviewed, only two were deemed to be sufficiently supported by high-quality evidence. The majority of the guidance documents had conducted a literature review to pinpoint evidence upon which recommendations could be built, however these were narrative literature reviews rather than comprehensive systematic reviews. Meanwhile, only two of the reviewed guidelines included a grading of the certainty or quality of the supporting evidence and only one of them had clearly linked the recommendation to the body of evidence.

In addition, most guidelines did not include evidence that included children with SHCNs. It should not be implied that recommendations ought not be made when there is no high-quality evidence relevant to the population of interest; however, the applicability of evidence extrapolated from other populations/settings should be discussed. This transparency for any extrapolation of evidence is vital, particularly for end-users of clinical practice guidelines implementing recommendations labeled as “evidence-based” [35].

To conclude, we discovered that very few guidelines had a clear link between evidence and final recommendation and very few applied clear grades to demonstrate the quality of evidence referred to. When rendering evidence-based recommendations more open, it would be advisable for evidence-rating frameworks to be more broadly applied (such as the GRADE system) [41]. This would need to go further than simply measuring the validity of the supporting evidence but also take into account the clinical relevance of certain evidence in the given situation. A detailed provision of the strengths and limitations of the evidence underpinning recommendations will thus enable clinicians to tailor the way recommendations are applied to their patients. Accordingly, the quality of research, the consistency of findings, the lack of ambiguity in the evidence, and the suitability of research design must all be taken into careful account [35].

In general, the most recent guidelines [33] received higher AGREE II scores compared with the other guidelines, which might indicate an improvement in the development of guidelines over the time period. This improvement should be applauded and encouraged, especially when a large organization such as the American Academy of Pediatric Dentistry (AAPD) acknowledges the importance of adhering to the recommendations of the Appraisal of Guidelines Research and Evaluation (AGREE) and other standards to improve the overall quality of clinical guidelines. These changes in the reporting and development of guidelines should be continued and this exemplary work should be followed by all developers for any future work.

The quality of clinical guidelines has been defined as “the confidence that the potential biases of guideline development have been addressed adequately and that the recommendations are both internally and externally valid, and are feasible for practice” [22]. The ratings of the assessors indicate that many of the guidelines and recommendations on oral healthcare for children with SHCNs have significant shortcomings, as assessed by the AGREE II tool. This can be best illustrated by the fact that all but one of the included (clinical) guidelines were recommended by the three assessors [33].

This study aimed to present a closer analysis of the existing state of the clinical guidelines in this specific area, which considers children with SHCNs and determines how much attention is provided to this particular group in the area of oral healthcare and dental treatment. The quality and reporting of the included guidelines for children with SHCNs in pediatric dentistry are very poor. Despite the existing number of guidelines and recommendations for children with SHCNs in dentistry, the current situation is not acceptable. The scientific community must come together to ensure that all children with SHCNs are treated optimally and equally.

## 5. Conclusions

The development and content of clinical guidelines in the area are concerning in terms of the low score attributed to the majority of the domains. Therefore, the scientific community must recognize the urgent need to enhance the quality and strength of the existing clinical guidelines on oral healthcare for children with SHCNs in the field of pediatric dentistry by the use, as well as the implementation, of current Clinical Practice Guideline reporting standards. Evidence-based practice has become very popular and is widely adopted across many disciplines. It will be advantageous for all stakeholders to work collaboratively to develop a comprehensive and meticulous evidence base that can be used to build clinical guidelines that are directly applicable to children with SHCNs.

## Figures and Tables

**Figure 1 ijerph-20-01686-f001:**
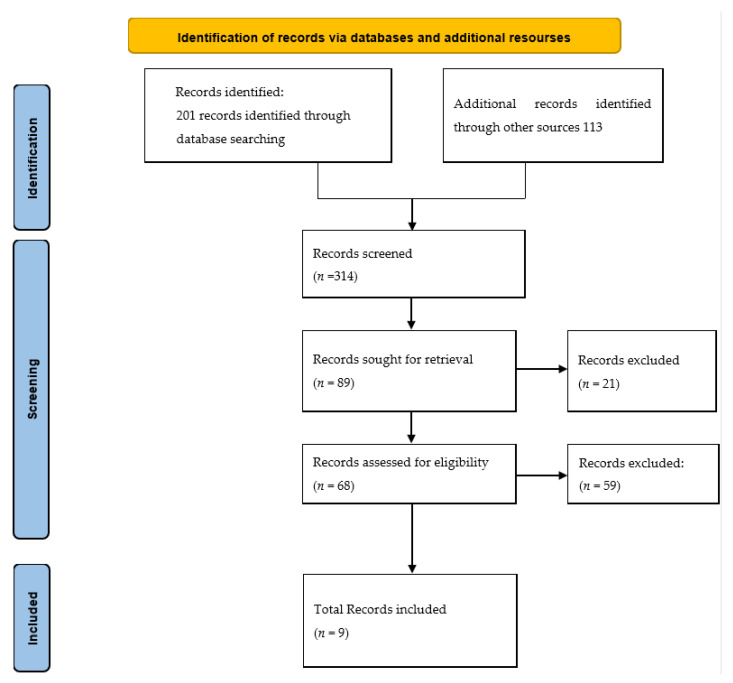
PRISMA flow diagram.

**Table 1 ijerph-20-01686-t001:** AGREE II domains and key items.

**1.1 Purpose and Scope**	This domain is concerned with the guideline’s overall aim, the target population, and the health-related questions (items 1–3).
**2.1 Stakeholder Involvement**	This domain lays focus on the extent or degree to which the guideline has been developed or formulated by the appropriate or right stakeholders and denotes the opinions of the intended users of the guideline (items 4–6).
**3.1 Rigor of Guideline Development**	This domain relates to the process that is used to gather, as well as produce, the evidence; the various methods that are used for formulating or developing the recommendations; to update or apprise them (items 7–14).
**4.1 Presentation of Clarity and Transparency**	This domain deals with the guideline’s format, structure, and language (items 15–17).
**5.1 Applicability of Guidelines**	This domain relates to the possible facilitators and barriers to implementation, approaches to enhance uptake or acceptance, and resource implications and consequences of guideline application (items 18–21).
**6.1 Editorial Independence**	This domain focuses on the formulation or development of recommendations without any bias (items 22–23).

**Table 2 ijerph-20-01686-t002:** The overall scores for each included guideline by the AGREE II domain.

Guideline	Scope and Purpose (%)	Stakeholder Involvement (%)	Rigor of Development (%)	Clarity of Presentation (%)	Applicability (%)	Editorial Independence (%)	Overall Assessment (median) (%)	Would you Recommend? (%)
Guideline on behavior guidance for the pediatric dental patient [25]	67	31	36	56	19	0	4-4	N, N, N
Clinical holding skills for dental services [26]	80	28	3	35	15	0	3-4	N, N, N
Clinical guidelines and integrated care pathways for the oral healthcare of people with learning disabilities [27]	81	83	59	69	31	0	4-4	N, N, N
Guideline on use of anesthesia personnel in the administration of office-based deep sedation/general anesthesia to the pediatric dental patient [28]	80	28	29	57	10	0	3-3	N, N, N
Guideline on management of dental patients with special healthcare needs [29]	85	28	24	67	15	0	3-4	N, N, N
Guidelines for monitoring and management of pediatric patients before, during, and after sedation for diagnostic and therapeutic procedures [30]	74	17	10	43	7	0	3-3	N, N, N
Guideline on use of nitrous oxide for pediatric dental patients [31]	78	26	28	63	17	0	3-3	N, N, N
Guideline on prescribing dental radiographs for infants, children, adolescents, and persons with special healthcare needs [32]	81	24	22	69	0	0	3-4	N, N, N
Use of silver diamine fluoride for dental caries management in children and adolescents, including those with special healthcare needs [33]	85	72	88	85	60	0	6-6	Y,Y,Y

The overall assessment is the median of the three appraisers; score of one indicates the lowest possible score to seven, the highest possible score. Y—yes; Y+—yes with modifications; N—no.

**Table 3 ijerph-20-01686-t003:** Summary scores for each domain by the AGREE II tool.

Domain	Median Score (%)	Range of Scores (%)
Scope and Purpose	80	67 to 85
Stakeholder Involvement	28	17 to 83
Rigor of Development	28	3 to 88
Clarity of Presentation	63	35 to 85
Applicability	15	0 to 60
Editorial Independence	0	0 to 0

## Data Availability

Not applicable.

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
