# Peer review of "Quality of Clinical Guidelines on Oral Care for Children with Special Healthcare Needs: A Systematic Review"

_ijerph, 2023, doi:10.3390/ijerph20031686_

Round 1

Reviewer 1 Report (Previous Reviewer 2)

Dear Authors!

Thank you for the answers, the manuscript imporved a lot.

Author Response

Dear reviewer,

I’m very much appreciated the encouraging, critical, and constructive suggestions and comments on this manuscript.

Comment 1:

Dear Authors!

Thank you for the answers, the manuscript improved a lot.

Response:

Thank you so much for your time reviewing our manuscript.

Reviewer 2 Report (Previous Reviewer 3)

The manuscript has been significantly improved and now warrants publication in IJERPH

Author Response

Dear reviewer,

I’m very much appreciated the encouraging, critical, and constructive suggestions and comments on this manuscript.

Comment :

The manuscript has been significantly improved and now warrants publication in IJERPH

Response:

Thank you so much for taking the time to review our work.

Reviewer 3 Report (Previous Reviewer 4)

Dear authors,

Thank you very much for your paper. In this paper, the authors presented a study entitled “Quality of Clinical guidelines on oral care for children with special healthcare needs : A systematic Review aiming to provide the most recent guidelines on oral health care management in children with special healthcare needs.  In general, the manuscript is very interesting.

  However, several corrections are required to improve the overall quality. An English-language review is required.

My recommendations are the following:

Unfortunately “Oral health conditions for children with special healthcare needs (SHCNs) are often  poorer when compared to healthy children” and this fact depends on several factors. Several studies have analyzed these factors, and among the most frequently encountered problems are unfortunately those related to architectural barriers. Please consider: https://doi.org/10.3390/ijerph18041556 , 10.3390/ijerph16152753

Please consider a flow chart in the materials and methods section with the included studies.  Have the authors considered the standards and guidelines given in the PRISMA Statement?

Add limitations of the study in a separate paragraph

Author Response

Dear reviewer,

We’re very much appreciated the encouraging, critical, and constructive suggestions and comments on this manuscript. The suggestions have been very thorough and useful in improving the manuscript. We have taken them fully into account in revision and the manuscript has been revised as per the suggestions given and our responses are as follows:

Suggestion 1:

Unfortunately “Oral health conditions for children with special healthcare needs (SHCNs) are often  poorer when compared to healthy children” and this fact depends on several factors. Several studies have analyzed these factors, and among the most frequently encountered problems are unfortunately those related to architectural barriers. Please consider: https://doi.org/10.3390/ijerph18041556 , 10.3390/ijerph16152753

Response- Thank you so much for your valuable comments and suggestion. We agree with your valuable suggestion and references were updated accordingly. 

Suggestion 2:

Please consider a flow chart in the materials and methods section with the included studies.  Have the authors considered the standards and guidelines given in the PRISMA Statement?

Response - Thank you so much for your valuable suggestion. A flow chart was added to manuscript (line 180-216).

Suggestion 3

Add limitations of the study in a separate paragraph

Response- Thank you so much for your valuable suggestion. Limitations were added in a separate paragraph line 290-300.

Thank you so much for taking the time to review our work.

Round 2

Reviewer 3 Report (Previous Reviewer 4)

The authors have greatly improved the quality of the manuscript and in my opinion it can be published. However, the bibliography should be formatted according to the journal guidelines.

Author Response

Dear reviewer,

We’re very much appreciated the encouraging, critical, and constructive suggestions and comments on this manuscript. The suggestions have been very thorough and useful in improving the manuscript. We have taken them fully into account in revision and the manuscript has been revised as per the suggestions given and our responses are as follows:

Suggestion 1:

The authors have greatly improved the quality of the manuscript and in my opinion it can be published. However, the bibliography should be formatted according to the journal guidelines.

Response- Thank you so much for your valuable comments and suggestion. The bibliography were formatted and updated accordingly. 

Thank you so much for taking the time to review our work.

This manuscript is a resubmission of an earlier submission. The following is a list of the peer review reports and author responses from that submission.

Round 1

Reviewer 1 Report

Review comments

This manuscript aimed to “critically appraise the most recent guidelines on oral health care management for children with special healthcare needs (SHCNs) in a paediatric dentistry setting” via the AGREE II tool. The authors concluded that “the current state of guidelines on oral care management for children with special healthcare needs (SHCNs) is, on the whole, of very low quality.”.

This review paper focused on an important topic of appraising the clinical guidelines. The methodology of conducting the systematic review was generally fine. However, some mechanical issues should be corrected before it becomes qualified to be published.

Introduction

1.     The authors mentioned the importance of guideline on oral health for SCHNs and the importance of appraisal the guideline. However, there was lack of demonstrating the research gap after the description (e.g, any guidelines on oral health for SCHNs existing by now? If yes, did they critically appraisal before releasing?), then presenting the aims/objective of the study. Please revise accordingly.

2.     Move texts on Line 84-89, Page 2 (aims of the study) to the last part of “Introduction”.

Materials and Methods

3.     The inclusion criteria should include the date restriction of the published literatures.

4.     The best practice suggest that two separate reviewers are needed to perform literature searching and sifting with a statistical test of inter-rater agreement, why there is only one rater in the screening? (Page 3, Line 110-113).

Results

5.     Revise the result part in accordance with the revision on the “Materials and Methods” part.

Author Response

Dear reviewer,

I’m very much appreciated the encouraging, critical and constructive suggestions and comments on this manuscript. The suggestions have been very thorough and useful in improving the manuscript. We have taken them fully into account in revision and the manuscript has been revised as per the suggestions given and my responses are attached. Thank you 

Reviewer 2 Report

Thank you very much for this interesting paper. This is a very current and forward-looking topic nowadays.

But there are points to consider for improving the manuscript:

The number of the references is too low and they are old publications. Without new and enough references you can not support your study.

Why did you choose AGREE II and not the usual way of systematic review or meta-analyzis? I missed the risk of bias and the statistical analyzis.

Overall, after fixing the problems, this can be a valuable manuscript.

Author Response

(The authors gave the same response as above.)

Reviewer 3 Report

Authors have chosen a good and current topic.

I would change the title to: Evaluation of existing guidelines....”

I would also write at the end: Systematic Review.

There are books, websites, professional materials, guidelines that were not published in scientific journals, but are still useful, current, and instructive.

https://ohd.moh.gov.my/images/pdf/xtvtnsop/Oral-Healthcare-for-Children-with-Special-Needs-Guidelines-for-Implementation-2004.pdf

https://www.aapd.org/research/oral-health-policies--recommendations/

https://link.springer.com/book/10.1007/978-3-030-10483-2

and many more besides.

It would have been worth considering these as well, not just the 9 guidelines and the 41 recommendations.

The thesis is much more valuable than the authors give it credit for. In the Conclusions chapter, it is only established that the guidelines need to be developed.

 I recommend the article for publication, but I would insert an Aim of study chapter, where specific objectives are listed. In the Conclusion section, a more specific critique and goals should be displayed, so I recommend a revision in this direction.

Author Response

Dear reviewer,

I’m very much appreciated the encouraging, critical and constructive suggestions and comments on this manuscript. The suggestions have been very thorough and useful in improving the manuscript. We have taken them fully into account in revision and the manuscript has been revised as per the suggestions given and our responses are attached. Thank you 

Reviewer 4 Report

Dear Authors,

I think this is a relevant review, with detailed methodology and interesting results. Good goals, well planned, well developed. Paper is clearly written, easy to understand concepts and statements. I advise publication after minor revision.

Here are my comments for revision:

The abstract and the introduction section are well-done .  Please specify the type of the manuscript(in your case a review) in the title and in the abstract.

The materials and methods section  is clear .

Line 2: “Specifically, children with SHCNs face many everyday challenges or problems in maintaining good oral health”. Please discuss more fully(in this section or maybe in the discussion)  the issues that affect people with disabilities. (i.e. presence of architectural barriers). 

Author Response

(The authors gave the same response as above.)
